# Application and Future Prospective of Lactic Acid Bacteria as Natural Additives for Silage Production—A Review



**Ilavenil Soundharrajan** [1], **Hyung Soo Park** [1], **Sathya Rengasamy** [2], **Ravikumar Sivanesan** [3] and **Ki Choon Choi** [1,*]

[1] Grassland and Forages Division, National Institute of Animal Science, Rural Development Administration, Cheonan 31000, Korea; ilavenil@korea.kr (I.S.); anpark69@korea.kr (H.S.P.)
[2] Centre for Research and Development PRIST University, Thanjavur 613-403, India; swisathya@gmail.com
[3] Department of Zoology, Rajah Serfoji Government College (Autonomous), Thanjavur 613-005, India; drravinikesh@yahoo.co.in
[*] Correspondence: choiwh@korea.kr; Tel.: +82-41-580-6752; Fax: +82-41-580-6779

**Abstract:** Ensiling is one of the essential processes to preserve fodder with high nutrients and microbiological quality. The forages before ensiling have a limited number of bacteria associated with the controlled fermentation process. Undesirable microbes can grow in silages when there is not efficient fermentation. Such kinds of microbes might cause pathogenic or toxic compounds that affect animal and human health. Therefore, it is necessary to inoculate potent starter cultures. Lactic acid bacteria's (LABs) have been considered the most prominent microbial additives used to improve the quality of silage. Currently, LABs have been used in modern and sustainable agriculture systems due to their biological potential. Recently, many scientists have increased their focus on developing nutrient-rich animal feed from forages with LAB. This current review focuses on issues related to forage preservation in the form of silages, how undesirable microbes affect the fermentation process, the critical role of LAB in silage production, and the selection of potent LABs to effectively control unwanted microbial growth and promote those which favor animal growth.

**Keywords:** ensiling; forages; fermentation; LAB; animal and human health

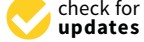



## 1. Introduction

Ensiling is a method to preserve raw plant materials based on spontaneous lactic acid production by controlling fermentation under anaerobic conditions. It has been used for many decades for the preservation of silages produced from various legumes, fodder, and residue crops (Figure 1). Forage preservation by the ensiling method has attracted great attention, providing consistent, reliable, and predictable feed supply for ruminant production. Risk of digestible nutrient losses by plant oxidation, undesirable microbial population in plants, proteolytic activity, *Clostridia* fermentation, microbial deamination, and decarboxylation of amino acids could negatively affect conservation efficiency and increase energy and nutrient losses as well as the accumulation of anti-nutritional compounds in forages [1]. Epiphytic lactic acid bacteria (LAB) utilize water-soluble carbohydrates present in ensiled plants and metabolize them into lactic acid, with a lesser extent of acetic acid which can lower the pH of the silage and prevent undesirable microbial growth, allowing them to be stored for a long time. An abundance of epiphytic bacteria in ensiled plant materials is not sufficient to induce the production of sufficient amounts of lactic acid in silage samples. Populations of LAB in plants are often heterofermentative and low in numbers [2]. Heterofermentative bacteria can increase the converting ratio of lactic acid into other metabolites such as acetic acid and ethanol. However, homo-fermentative bacteria do not convert lactic acid to other organic acids. Thus, a high level of lactic acid is sustained in the silage. In general, lactic acid found at the highest concentration is an indicator of good silage [3]. To make high-quality silage with strong digestibility, stimulation of the ensiling process is required by adding different types of chemical and biological additives.

Currently, the use of additives is recommended to ensile green folder with significant concentrations of mono-, di-, and oligo-saccharides and high protein content with high buffer capacity. Expected changes in silage production when ensiled with LAB include an increased ratio of lactic acid with marginal amounts of acetic acid, reduced proteolysis, and increased dry matter recovery [4].

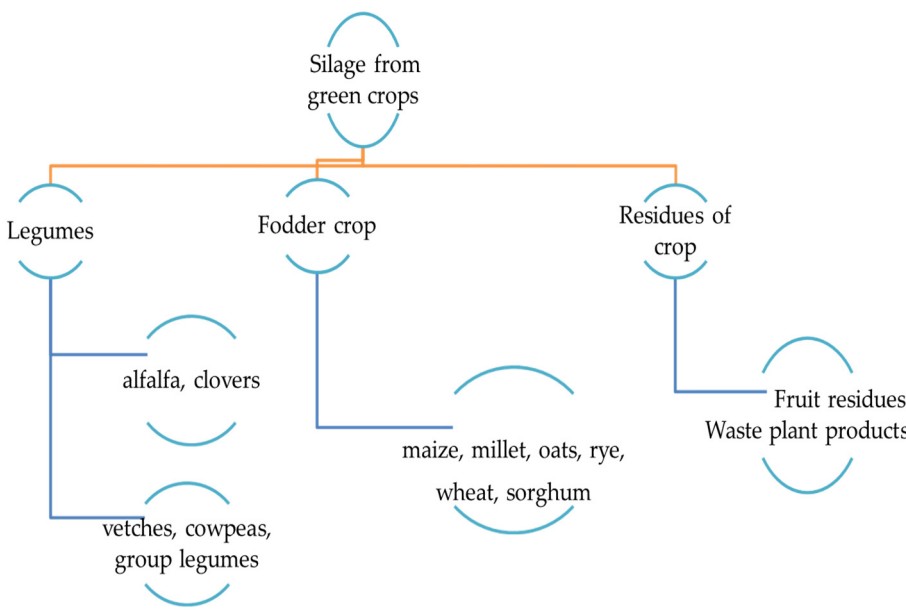

**Figure 1.** Sources of silages production from various crops.

## 2. Features and the Procedure in Producing Silage

Ensiling is an alternative and essential technique to preserve silage for a long time. Silage has been produced from grass and other green fodders and fermented by naturally occurring microbes under anaerobic conditions. The process requires consideration of a broad range of different factors such as plant growth, harvest, storage, and feed practices (Figure 2). Considering the desirable final products will help us to determine the necessary steps in silage production. In general, the major objectives are to preserve the digestible fiber, protein content, and energy in the forages, which can be utilized efficiently by ruminants. Choosing an exact time to harvest (ideal phase of maturity) the forage plants could directly influence the quality of forage to be preserved as silage. Instead, when harvesting the plants at the mature stage (for higher yield purposes) reduces the quality of silage. Field chopping, transporting forage, and filling the silo are the key components for the rapid harvesting of forage crops. The quality silages are based on timely and uninterrupted movement from the field to the silo. The forage may be cut with the sickle bar or drum mowers and will be allowed to wilt in windrows in the field. Developing silage includes cutting or shearing processes. The forage cut is important; the specified length is 6–60 mm and looks sheared for better packing with LAB and air extraction during the ensiling process. Fresh forages have higher moisture content (>80%), soluble proteins, and sugars in the liquid that become more suitable for molds, yeasts, and bacteria; and the enzyme activity is also more active in the liquid. Therefore, we realize such activity of microbes and enzyme is important during silage production. The good ensiling process creates an oxygen-free environment to stimulate lactic acid bacteria while inhibiting yeast, mold, and other undesirable microbes. During this time, the naturally occurring microbes can change water-soluble carbohydrates into lactic acid, a major acid present in fermented silages, and lower the pH of the silage. Lactic acid content indicates the quality of the silage [3]. The most important purpose of ensiling is to preserve forage accessibility for a whole year without any damages, thus improving the economic and environmental sustainability of silage production.

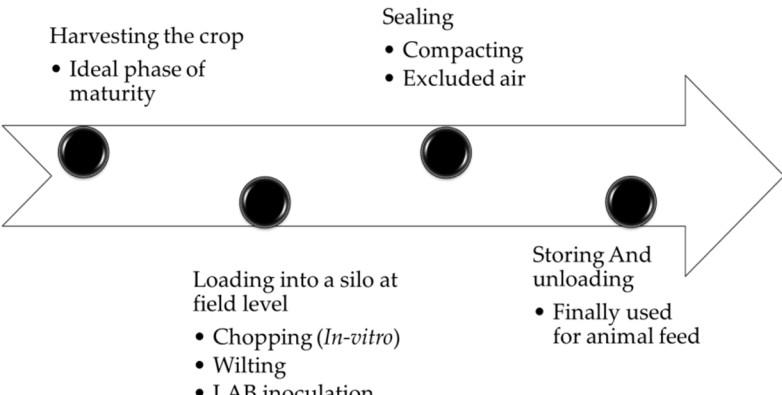

**Figure 2.** Stages of silage production from different plants.

### 3. What Happens to Samples When They Are Ensiled at Different Stages?

The researchers divided the fermentation process of silages into different phases (Figure 3), including the initial aerobic phase, fermentation phase, stable phase, and aerobic feed-out phase [5,6]. The aerobic phase starts from the movement of substrates to inside the silo via the compaction process until the oxygen levels are reduced [5]. This phase has the most notable features, such as increased temperature (>32°) of samples due to cellular respiration by plants and aerobic microbes. In this condition, the amount of energy generated by respiration is much higher, and this energy can be dissipated in the form of heat. With increases in acidification and concentrations of oxygen, the microorganisms that cannot survive in these conditions are inhibited, but those microbes can have survival properties that succeed. Epiphytic/endophytic microorganisms of plants, possible contaminants, or manually added inoculums are present in this stage. The most common epiphytic microbes are enterobacteria [7], yeast, molds [8], and low numbers of LAB [9,10] with considerable diversity among plants, climates, and forage crop management.

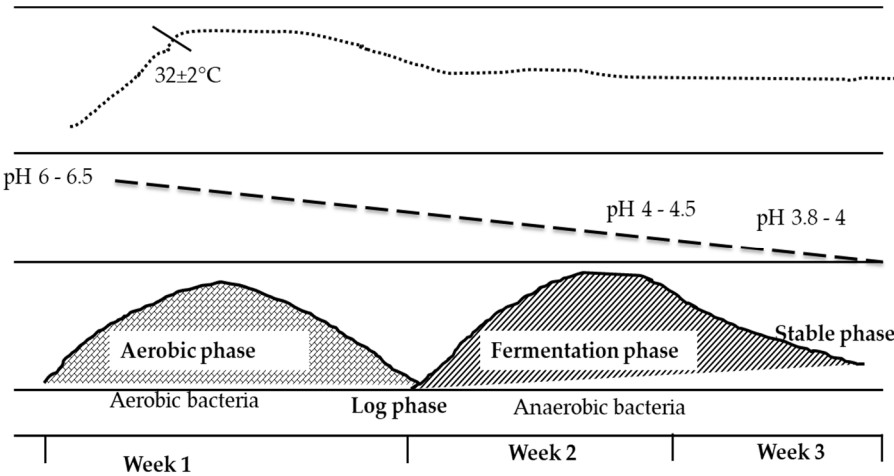

**Figure 3.** Phases of silage fermentation at ensiled plant materials.

Fermentation phase: in this stage, the fermentation process is accelerated by the dominant microorganism. In well-compacted silages with sufficient water-soluble carbohydrates, the LAB can dominate the fermentation process due to their adaptive and competitive growth characteristics, resulting in well-preserved materials. The population and epiphytic diversity of LAB is highly variable and interferes with the process. In general, if LAB dominates the process, lactic acid fermentation has been started; however, it may be varied by the proportion of homofermentative or heterofermentative fermentation as a result of variations in the lactic acid to acetic acid ratio from 1:1 to 6:1 [3]. The duration of

this phase, or intensity of fermentation, highly depends on the availability of substrates and growth conditions for dominant microorganisms. If not, LAB dominates, and a failure to accelerate the lactic acid fermentation and reduction in pH (pH 4.0–4.5) results in undesirable microbes such as *Clostridium*, *Listeria* spp., and propionic acid bacteria (PAB) starting to grow and affect the silage quality [11]. Soundharrajan et al. studied the impact of LAB on the fermentation of Italian ryegrass at different storage periods. Reduction in pH and higher lactic acid content were noted in silage treated with LAB at 45 d and 95 d. Meanwhile, the further extent of storage periods affects lactic acid fermentation due to lower LAB at 180 d and 360 d, resulting in lower lactic acid and higher acetic acid [12]. This may be due to nutrient availability of the forage samples, which favor growth of the other microorganisms. In another study [13], rice straw treated with LAB had almost similar lactic acid and acetic acid contents throughout the experimental periods (45 d, 90 d, and 365 d). This confirmed that the fermentation process is highly based on the availability of substrates in plants. In the stable phase, the reactions are minimal, and the fermentation characteristics of the silages are affected by storage periods. During this phase, the pH of the silage is between 3.8 and 4.0. The storage time affects the pH, lactic acid, acetic acid, propionic acid, and butyric acid, 1, 2-propanediol, the lactic/acetic acid ratio, and NH3-N [12,14]. All chemical and microbiological properties of the silage in the stability phase will affect the intensity of the deterioration after the silo opens. When the silo opens for the feed-out (feed-out phase), the fermented silages are in contact with air, which alters the conservation principles, anaerobiosis, and environmental conditions [5,15]. In this stage, there are microbes that have survived in the active form or inactive spores whose growth is limited only by the oxygen availability that favors microbe growth. Among the microbes, initially, the yeast starts the process because they are facultative anaerobes, and most of the yeast has tolerance to acidity, followed by the growth of mold, aerobic bacteria (that reduce lactic acid), LAB populations, and increased pH [5,14]. These conditions can favor the growth of undesirable microorganisms that affect the silage quality.

## 4. Naturally Occurring Microbiota in Plant Samples

Plant materials, including grass and legumes, are intended for use as silages that have different types of aerobic and anaerobic microbes on their surfaces. They are called epiphytic microflora. The microflora levels are based on types of raw materials (grass and legumes) at different stages and environmental factors (temperature, soil, harvesting method, and agro-technical). Native bacteria greatly determine the fermentation of ensiled materials via different types and amounts of fermentative metabolite productions. At the same time, they affect the stability of obtained silages [16]. The abundance of microflora in plant parts consist of LAB and other undesirable microbes that influence the fermentation process and the quality of silages. Firmicutes and Proteobacteria are the most abundant phyla in ensiled silages [17,18]. Bacteria of the Enterobacteriaceae family, acetic acid bacteria, spore-forming bacteria (*Bacillus* and *Clostridium*), and *Listeria* are considered undesirable [5]. Several species of molds and yeasts are also considered undesirable microbes [19]. Epiphytic LAB plays a major role in spontaneous silage fermentation under anaerobic conditions. They are a relatively small group of microorganisms, not exceeding 1% of total microflora in plants. Dominant LAB involved in the fermentation of silages includes bacteria belonging to the genera *Lactobacillus*, *Pediococcous*, *Leuconostoc*, *Weissela*, and *Lactococcus* [20]. Common species belonging to the genus *Lactobacilli*, such as *Lactiplantibacillus plantarum*, *Levilactobacillus brevis*, *Lacticaseibacillus casei*, *Lacticaseibacillus rhamnosus*, *Latilactobacillus curvatus*, *Lactobacillus gasseri*, and *Lactiplantibacillus pentosus*, as well as species belonging to the genus of *Pediococcus*, such as *P. pentosaceus*, *P. acidilactici*, and *P. damnosus* have received great attention for preserve silage quality with high nutrients. Other microflorae such as *Bacillus megaterium*, *Weissella kimchii*, *Enterococcus flavescens*, *Lactobacillus taiwanensis*, *P. lolii*, *Leuconostoc lactis*, *E. mundtii*, and *W. cibaria* are also involved in the fermentation of silages [11]. However, the effects of these inoculants on silage production can vary based on the crops used [1].

## 5. Beneficial Microbiota and Their Impact on Ensiled Silages

Various types of microbial populations have been found in plant samples. They play major roles in silage fermentation. Beneficial microbes such as *B. megaterium, W. kimchi, P. lolli, L. lactis, E. mundtii, W. cibaria, W. flavescens, L. plantarum,* and *L. rhamnosus* are commonly found in maize, rice, sorghum, and alfalfa silages [21,22]. Epiphytic microflora can significantly regulate the fermentation of ensiled materials by affecting the amount and type of organic acids produced [5]. Constituents and accumulations of epiphytic microflora on ensiled plant raw materials are inadequate to initiate the production of lactic acid. Natural quantities of LAB on plant materials are often heterofermentative with low numbers [2,12]. Robust species involved in silage-making processes are mostly *Lactobacillus* and their sub-species such as *L. brevis, L. casei, L. rhamnosus, L. curvatus, L. gasseri,* and *L. pentosus* and *Pediococcus's* such as *P. pentosaceus, P. acidilactici,* and *P. damnosus,.* However, levels of both genera do not exceed $1 \times 10^3$ CFU [23]. A novel *L. taiwanensis* species has also been isolated from silages [24].

## 6. Undesirable Microbial Population and Its Complication in Ensiled Silages

A group of microorganisms (bacteria, yeast, and mold) can have a negative impact on the silage-making process that can decrease the silage quality [25]. Bacterial flora (*Clostridium, Bacillus, E. coli, Enterobacter, coliform bacilli, Listeria, Salmonella, E. faecium, E. faecalis, E. mundtii, E. casseliflavus, E. avium, E. hirae*) can cause undesirable effects during the fermentation process [26]. Most mold species are proficient in producing secondary metabolites such as mycotoxins which are toxic to humans and animals, the occurrence of these toxins in feed can endanger food safety [27]. Silages can be contaminated by toxins produced by molds belonging to the genus *Penicillium,* such as *P. verrucosum. Aspergillus* species (*A. ochraceus, A. sulphureus*) can produce ochratoxin and aflatoxin [28]. Microflora of silages prepared from maize shows the most toxic molds such as *Arthrinium phaeospermum, Byssochlamys* sp., *Fusarium* sp., *Monascus ruber,* and *Penicillium* sp. [29].

### 6.1. Yeasts and Molds

Yeasts and molds can rapidly degrade the silage quality. They are involved in the aerobic phase at the beginning of ensiling or during the acceptance phase (they tolerate acidic conditions) to regenerate organic acid during the metabolic process [30]. Processed silages can be affected by many genera of fungi (fungi are aerobic microbes found outside of the silo, in the unfolding phase). Brazilian research studies have shown that nearly 195 strains are found in corn silage samples tailed by *Fusarium, Penicillium, Aspergillus, Trichosporon* sp., and *Cladosporium* sp. [31]. Generally, fungi can produce numerous toxic secondary metabolites (mycotoxins). Most fungi remain after processing (*Penicillium* 70%, *Fusarium* 47%, and *Aspergillus* 34%). Mycotoxin-producing fungi are more frequently isolated from corn silages [32]. Fusarium species can produce 20 mycotoxins (deoxynivalenol, zearalenone, fumonisin) [33]. Nearly 300 mycotoxins have been comprehensively studied in corn silages (only a few strains are responsible for food safety). They can cause spoilage (aflatoxin). Aerobic exposure can cause spoilage during the initial silage-making process [34,35]. Chronic exposure of mycotoxin to silage can contaminate animal feed and cause non-specific symptoms (hormonal inequality, immune system damage, metabolic damage). Cattle, poultry, and sheep are resistant to mycotoxin (deoxynivalenol). However, mycotoxin can decrease feed intake (10–20 mg/kg) of ruminants [36].

Recent studies have proposed that beef cattle become affected by Jejunal Hemorrhagic Syndrome (JHS) after consuming feed containing mycotoxin-producing fungi (*Fusarium, Penicillium, Aspergillus* sp.) [37]. Transmission of toxins to dairy and meat goods is a possible risk for humans. Ruminants are better protected from several mycotoxins than other animals due to the biotransformation ability of certain rumen microorganisms such as ciliate protozoa (e.g., ochratoxin A produced by *A. ochraceus* and *P. verrucosum*). However, these mycotoxins are nephrotoxic to many animals [38]. High-producing silage ruminants are normally fed on silage-based diets with improved levels of concentrates. The ensuring

acidification of the rumen environment makes animals sensitive to mycotoxins, perhaps through improved absorption and reduced microbial detoxification [39].

### 6.2. Bacterial Species

Accumulation of undesirable bacteria in silages is usually linked to the onset of a quick illness. The infection affects both humans and animals through direct contact between bacteria (e.g., *Listeria monocytogenes*) and hosts and the synthesis of biogenic amines (toxins). Butyric acid bacteria (BAB) can be transferred from grass and corn silages to animals [40]. BAB is an endospore-forming bacteria; it includes several *Clostridium* strains (*C. tyrobutyricum, C. butyricum*). These endospore-forming bacteria have been found in the GI tracts of dairy cows [41]. *B. cereus* species are present in milk, and their products are considered a major source of food poisoning [42]. The existence of *Clostridium* species in milk can cause off-flavor and excessive gas production in hard/semi-hard ripened cheeses. The spoilage affects both the smoothness and taste of ripened cheese [43]. *C. botulinum* is known to produce an extremely pathogenic toxin that frequently causes the death of animals and humans. *C. botulinum* is the most dominant pathogenic bacterium present in poorly made silages; it can cause botulism. This bacterium can be transferred from spoiled milk and farm atmospheres to humans and animals via the dairy chain [44]. The source of endospores or toxins is mostly moist silages, acidified silages, and animal carcasses that promote toxin synthesis [45]. *Listeria* spp. as pathogenic bacteria are present in various environments, including silage, grass, water, and so on [46].

The dominant species *Listeria innocua* and *L. ivanovii* are major causes of animal infections. *L. monocytogenes* present in silage can cause listeriosis in a number of animals and humans [47]. Listeria species are mostly present in milk. They can be easily transferred to humans, causing infections [48]. The population of *L. monocytogenes* (6%) found in corn and grass silages at pH 4.5 increases the *Listeria* species [49]. Animal feed silage is also considered a vector for spreading pathogenic *E. coli*. The prevalence of *E. coli* O157: H7 has been found in corn silages [50]. The occurrence of *E. coli* O157 is only found in very poor silage [51]. The silage-making process at reduced pH levels can influence the growth of pathogenic *E. coli*. Increased pH can influence shiga toxin-producing *E. coli* strain's surveillance [52]. During the silo opening stage, dangerous contamination can occur due to mycotoxin and shiga toxin-producing strains and cause multiple disease complexes, including hemorrhagic syndrome in beef cattle [53]. The most contagious disease in wildlife and domestic cattle is infection by *Mycobacterium bovis* that can be easily transmitted to domestic animals through close contact [54]. The presence of *Yersinia enterocolitica* has been found in silages (46 silage samples). Nearly 6.5% of Yersinia pathogens are associated with various kinds of silages. These pathogens can cause a disease called yersiniosis (zoonotic) in humans. This disease can also occur in a wide range of animals such as cattle, pigs, deer, and so on (Center for disease control and prevention, 2005). Occasionally, silage food-borne infections caused by *Campylobacter sp.* can occur [55]. The European Commission reported a list of pathogenic microbes and screened animals before thrashing. *Salmonella* responsible for diarrhea is usually present in cattle with a prevalence of 0–90% depending on the animals and geographic regions [56].

### 6.3. Bacterial Synthesis of Biogenic Amines

Most bacteria can synthesize biogenic amines (BA) during decarboxylation. Many strains (e.g., *Clostridia, Bacillus, E. coli,* and *Pseudomonas*) are present in silages. Current research has described that large quantities of tyramine, cadaverine, putrescine, and lower quantities of arginine, histamine, histidine, spermidine, and tryptamine are present in silages. These biogenic compounds have a number of negative effects on animals (ketonemia, histaminosis in ruminants). The biogenic amines (BA) can enhance proteolysis and are linked to nutritional markers of the silage. They can also reduce protein contents [57]. Feeding of silage containing 100 g putrescine per day can lead to anorexia in most cows. BA concentrations in the rumen can only be decreased by amine-degrading microorganisms.

Histamine-producing bacteria can play a vital role in bovine laminitis (present rumen of cattle). BA formation is affected by pH and temperature at the initial stage of infection [58]. Surveillance of biogenic compounds such as putrescine (136 mg/kg), tyramine (145 mg/kg), cadaverine (96.2 mg/kg), spermidine (37.9 mg/kg), histamine (3 mg/kg), spermine (2.8 mg/kg), and tryptamine (2.5 mg/kg) were found in the silage samples.

## 7. Synthetic and Natural Additives for Silage Production

### 7.1. Synthetic Additives for Silage Production

To maintain the quality of silage, several numbers of additives can be used to inhibit the growth of undesirable species. These inhibitors can be added during ensiling. For example, sodium nitrite and hexamine can prevent the growth of *Clostridia*. The growth of yeast can be restricted by sodium benzoate [59]. The use of chemical additives (calcium format, sodium nitrite, sodium benzoate) can improve the hygiene of corn silage and reduce concentrations of deoxynivalenol, zearalenone, fumonisins, and so on [60]. The addition of silage inoculants and organic acid can be used to synthesize some antimicrobial agents (ethanol, $H_2O_2$, exopolysaccharides, diacetyl) and antibacterial pesticides (bacteriocins).

### 7.2. Natural Additives for Silage Production

The current practice of silage-making combines several LAB strains [61–65] to induce silage fermentation (alfalfa, legumes, maize, grains, meadow, etc.) through a synergistic action, thus enhancing the stability for more than one year of storage [66]. Lactic acid bacteria have been used for several centuries for the production of feed, silages, and food. LAB has a potential role in decreasing the pH, thus offering protection against harmful microorganisms. LABs have a good impact on humans and animals because they can act as probiotics. Feed companies are very interested in using LAB inoculums for silage making because LAB can fight against pathogens and enhance the quality of silage with suitable parameters. Insight on the current silage fermentation process has been gained due to advanced molecular techniques, metagenomic, and novel techniques that can target inoculants for silage production [67]. The significance of LAB in recent years has been comprehensively studied using novel strains to improve the silage-making process [12,68,69]. Selections of microbial inoculants are considered to be very precious.

## 8. Changes in Fermented Silage by LAB

LABs are mainly responsible for pH reduction and preservation of nutrients in ensiled silages for a long time [11]. Microbiological additives are frequently used in silage fermentation [4,70]. Microbial inoculant is a mixture of one or more species of microorganisms that should be viable at the time of use. Microbiota in the silages has positive impacts (Figure 4) by decreasing dry matter losses, increasing essential metabolites of interest, inhibiting undesirable microbial growth, and improving nutritional quality [1,4,70]. Selecting potent microbes is essential to achieve positive effects. When selecting novel microorganisms, we should focus on specific characteristics of target substrates and general conditions of the ensiling environment to ensure optimal effects [68,71,72]. The addition of LAB during the ensiling process improved fermentation quality (higher lactic acid) and maintained the crude protein (CP), acid detergent fiber (ADF), and neutral detergent fiber contents (NDF) [12,13]. Another report claimed that the addition of LAB significantly increased nutritive profiles of silage at different storage periods [73]. Increased acid detergent lignin [74] and dry matter content (DM) [75] were noted in silages treated with LAB, but the DM level was varied at different storage periods [61]. The organic matter (OM) and DM content were increased in the fresh and rain-treated ryegrass silages treated with LAB [76]. The water-soluble carbohydrate (WSC) level was reduced in the silage treated with LAB [74,75]; this may be due to LAB being able to utilize WSC and convert into organic acids. The plant metabolites, such as 3-hydroxydecanoic acid, 2-hydroxy-4-methylpentanoic acid, benzoic acid, catechol, hydrocinnamic acid, salicylic acid, 3-phenyllactic acid, 4-hydroxybenzoic acid, (trans, trans)-3,4-dihydroxycyclohexane- 1-carboxylic acid, p-hydrocoumaric acid,

vanillic acid, azelaic acid, hydroferulic acid, p-coumaric acid, hydrocaffeic acid, ferulic acid, and caffeic acid were increased in grass silages inoculated with LAB [77]. LAB increased α-tocopherol levels in silages prepared from the mixture of birdsfoot trefoil and timothy, red clover and meadow fescue, or red clover and timothy; whereas, β-carotene levels were either the same or slightly reduced in same silage mixtures [78]; whereas, the silages from rye had higher β-carotene [79].

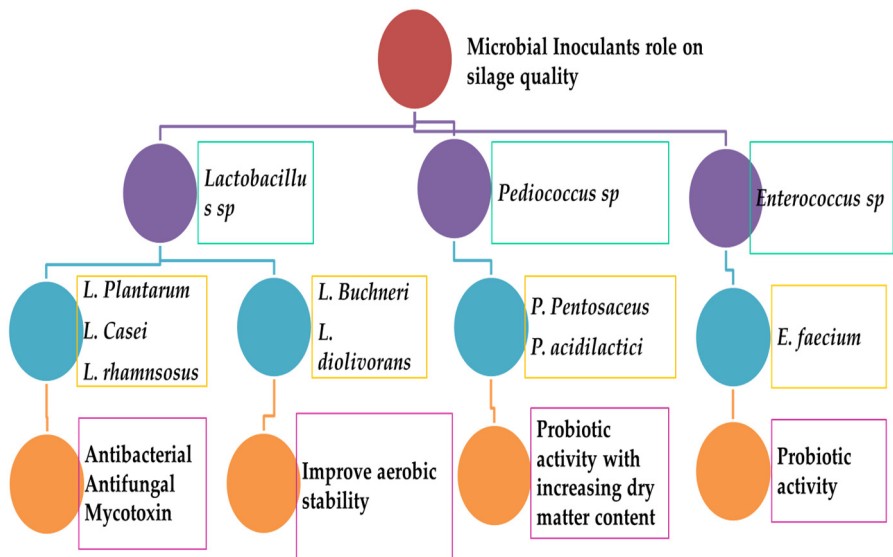

**Figure 4.** Role of lactic acid bacteria on silage production and preservation.

The current research practice focuses on homofermentative and heterofermentative lactic acid bacteria. Some non-LAB groups, chemicals, and enzymes are also involved in the silage-making process. Major factors that are important during silage fermentation include aerobic constancy, livestock consumption, and usage. The silage fermentation process involves two major microbes: homofermentative and heterofermentative species. Homofermentative inoculant groups include *Lactobacillus, Pediococcus,* and *Lactococcus* species. These inoculants can lead to the high productivity of lactic acid, lower the pH, and reduce the breakdown of proteins and sugar molecules in crops. Heterofermentative inoculants include *L. buchneri* and *L. brevis*. Both species can produce a mixture of lactic acid and acetic acid that can prevent the growth of yeast and mold as contaminants. Recent updates on homofermentative lactic acid bacteria have revealed that they are highly dominant during the fermentation of silage and can lead to the high quality of products. These homofermentative inoculants include *Lactiplantibacillus plantarum, Lactobacillus acidophilus, P. acidilactici, P. pentacaceus,* and *Enterococcus faecium*. The growth rate of bacterial inoculum on the dry matter content can increase very rapidly (*Enterococcus ≥ pediococcus ≥ Lactobacillus*). Most *Pediococcus* can tolerate higher DM content than *Lactobacillus* at a wide range of pH values and temperatures [80].

## 9. Insight on Homo- and Heterofermentative LAB

Homo- and hetero-fermentative LAB are widely used for the fermentation of silage. Recent updates on silage-making bacteria have been documented as facultative heterofermentative and relatively obligate homofermentative bacterial species [5]. Both species have diverged characteristics. Notably, homofermentative bacteria (ferment hexoses) and facultative heterofermentative bacteria (ferment pentoses) can both produce lactic acid. Facultative species include *L. plantarum, L. casei, Enterococcus faecium,* and *Pediococcus* sp. Higher lactic acid content is suitable for better recovery of the dry matter content of silages. A recent meta-analysis has revealed that the effects of various inoculants are different, depending on the specific kind of crop. Low pH and temperature can be maintained in

legume plants, alfalfa, and grasses. However, they are not maintained in other plants (corn, sorghum, sugarcane). The reduction of acid using culture inoculum plays an important role in maximizing crop and dry matter content recovery, reaching ≥2.8% for most grass silages. However, it is linked to losses (≤2.4%) for untreated silages (corn, sugarcane, sorghum) [1]. Heterofermentative LAB usually can synthesize a high volume of acetic acid during silage making, which can prevent fungal growth and consequently store silages longer under exposure to air. The microbial inoculum rate is usually $10^5$–$10^6$ cells per gram of crop [81]. Homo- and hetero-lactic acid bacteria (silage inoculants) are involved in alfalfa silage-making for the purpose of excellent animal feed production. A meta-analysis studied the effects of homo- and hetero- LAB on fermentation parameters, the value of nutrition, media composition, and aerobic stability of forages (alfalfa). Trending reports have suggested that both homo- and hetero-fermentative bacteria are good inoculants for enhancing the silage quality, reducing the contamination (yeast and molds), and increasing the forage conservation for livestock production [82].

### 9.1. Homofermentative LAB

The most commonly available starter culture contains homofermentative LABs, which are robust and efficient to produce lactic acid and thus improve the quality of silages. *Lactiplantibacillus plantarum, Lactobacillus acidophilus, E. faecium, P. acidilactici* and *P. pentosaceus* are the most popular LAB species [67]. Homofermentative bacterial groups are more powerful than heterofermentative ones. The significance of homofermentative bacteria is that they can catalyze each glucose molecule of lactic acid and yield high dry matter with less energy reduction for silages. Lactic acid is a more potent acid that can reduce silage pH more than other acids. The final pH is increased by a scale-up process of heterofermenters. Native bacterial populations are highly different across plant environment ecosystems. The addition of homofermentative bacterial inoculum can reduce the pH very fast while inhibiting other harmful bacterial contamination and storing plant proteins. The homofermentative inoculum can increase animal performance (3–5%). The most common microbial inoculants used for the production of silages are homofermentative LABs. The current scenario has shown that numerous bacteria are involved in the homo LAB fermentation process [16].

Animal trials have also shown the significance of homofermentative bacteria. They play a pivotal role in silage-making and milk production, can increase animal performance, inhibit the growth of most harmful bacteria, reduce toxin production, and result in good interactions with rumen microbes [83]. The encouraging effects of homofermentative microbes used in silage for reducing dry matter losses (DMLs) and improving the synthesis of metabolites are based on significant features of target substrates and suitable environmental parameters [84]. Generally, DMLs are linked to dominant kinds of metabolic activities with the tiny loss measured by the use of homofermentative bacteria [11]. These losses always depend on the DM concentration of the silage. High moisture content increases these losses. Homofermentative bacteria are considered good for reducing such losses. This parameter is linked to the potential of the inoculants. It depends on the strain and the substrate, thus influencing silage fermentation [85].

### 9.2. Heterofermentative LAB

Both facultative and obligate heterofermentative inoculums are involved in the silage-making process (*Lactobacillus, Oenococcus, Leuconostoc, Weissella*). The most dominant LABs are *Lentilactobacillus buchneri, Limosilactobacillus reuteri, Lacticaseibacillus casei* [86]. Other LAB groups include *Levilactobacillus zymae, Apilactobacillus kunkeei, Levilactobacillus acidifarinae, Levilactobacillus namurensis, Levilactobacillus brevis, Levilactobacillus spicheri, Fructilactobacillus fructivorans, Fructilactobacillus fructivorans,* and *Levilactobacillus hammesii* [87]. The obligate heterofermentative *L. buchneri* is considered the best silage additive. These bacteria can increase aerobic stability during heterofermentation. It can synthesize antifungal components [88]. A recent report has also suggested *L. buchneri* can strongly

produce antimicrobial compounds (salicylic acid, 3-phenyl lactic acid, catechol, benzoic acid, hydrocinnamic acid, 4 hydroxybensoic acid) during grass silage fermentation [77]. These obligate heterofermentative bacteria (*L. buchneri*) can be used as silage additives to induce aerobic stability during silage fermentation. They can lead to a medium level of acetic acid increase and reduce contamination of yeast [89]. Recent reports have revealed heterofermentative additives *Lentilactobacillus diolivorans* and *L. reuteri* of silage [90]. The most potential role of *L. buchneri* is that it can produce ferulic acid (esterase) in silage, leading to efficient fiber digestibility [91]. These heterofermentative bacteria can synthesize a sufficient amount of enzymes to produce suitable quantities of silages.

Microbial additives have been exploited for enriching the fermentation process of silage. Heterofermentative LAB strains can produce lactic acid and carbon dioxide. They also produce very limited amounts of by-products such as acetic acid and ethanol [92]. Heterofermentative bacterial inoculants are essential to reduce the spoilage of silage [93]. These bacterial inoculants are advantageous in that they can improve the quality of fermentation for the silage [66]. Heterofermentative LABs at 55 to 60 d can significantly increase aerobic stability [4]. Heterofermentative LAB (*Lentilactobacillus diolivorans*, and *Lentilactobacillus hilgardii)* strains have potential antimicrobial activities. They can inactivate pathogens after enzyme exposure [94]. *L. hilgardii* can synthesize a unique compound (phenyllactic and 4-hydroxyphenyllactic acid) that can fight against contaminants [95]. It has synergistic effects that increase aerobic stability and enhance prolonged fermentation of silages and other fermented foods [96] (S). It has been reported that *L. buchneri* can convert lactate to 1,2-propanediol, which does not usually take place ($\geq$ 21 d) during fermentation [97].

A combination of numerous strains has been used as microbial inoculum. The production also varies using potential groups of *L. plantarum, P. acidilactici,* and *P. pentosaceus* [98]. The paring of inoculants for 14 days of fermentation can give a successful production of corn silage [99]. A multi-inoculum preparation for alfalfa *L. buchneri* treated with *P. pentosaceus* for one week has resulted in a greater pH decline than untreated silage. A combination of *L. buchneri, L. plantarum*, and *L. casei* has been used for making barley silage on a lab scale [100]. The *P. pentosaceus, P. freudenreichii,* and *L. buchneri* are the most commercial inoculants used for bermudagrass silage production. The combination of more inoculants can enhance the initial stage of the fermentation process [101].

## 10. Criteria for Selection of LAB for Silage Production

The main reason for the production of silage fermentation using LAB is to have an excellent quality, a rapid reduction in pH, and improved biomass of silage [102]. Lactic acid bacteria play a role in the conversion of water-soluble carbohydrates into organic acids; it induces the acidification of silages and kills or prevents detrimental microbial growth. When selecting a potent LAB for silage production that LAB should have the following criteria; rapid growth (high cell density at ensiled silages), rapid reduction in pH, potent inhibition of pathogens (bacteria, yeast, and toxic metabolite producing mold), higher production of lactic acid with a marginal level of acetic acid, very low or complete inhibition of production of butyric acid content (a negative indicator of silages), increased aerobic stability and dry matter content (DM), and long-term preservation with rich nutrients [61,103,104].

## 11. Summary and Conclusions

Lactic acid bacteria (LAB) are considered potent natural additives for animal feed production due to the efficient production of biological metabolites—notably, higher lactic acid content with marginal level acetic acid and other organic acids. Furthermore, LAB can utilize water-soluble carbohydrates and convert them into valuable organic acids, which increase the acidification of the surrounding environment. Rapid acidification (lower pH) could help to prevent undesirable microbial growth and toxic secondary metabolite secretion. In addition, LABs have the GRAS (Generally recognized as safe) status. The traditional use of LAB and recent exploration of its knowledge about positive impacts

on animals and human health as probiotic potential provides hope for using LAB as an alternative tool for preserving both animal and human foods in agriculture sectors and food manufacturing companies. The inhibition of microbial pathogens is a primary criterion for producing and preserving silages long-term without loss of nutrients. Many researchers have experimented with different experimental models on the inhibitions of unfavorable bacteria, yeast, and mold growth, producing toxic metabolites. The study has suggested that LABs have potent antimicrobial activity by producing various organic acids and peptide-like bacteriocin. However, fermentation capabilities and inhibitions of pathogenic microbes have varied between the strains from either the homofermentative or heterofermentative groups of bacteria. Some other studies have recommended that mixed inoculants for silage production have a more significant impact on silage fermentation than the single culture use. The selection of starter cultures of LABs is characterized by the ability to reduce the undesirable microbial growth (*Clostridium*, *E. coli*, *Salmonella* spp., or *L. monocytogenes*) and might be an optimal tool for the production and preservation of green silages. Production of animal feed with LAB is not only inhibiting detrimental microbes but also enhancing beneficial microbes in the gastrointestinal tract, which helps to maintain animal health. This strategy would provide an excellent opportunity to make potential silages with better nutrient values than the harvested crops in the future.

**Author Contributions:** All authors contributed equally to this work. Conceptualization, I.S. and K.C.C.; writing—original draft preparation, I.S. and R.S.; writing—review and editing, H.S.P., I.S. and S.R.; visualization, H.S.P., K.C.C. and S.R. All authors have read and agreed to the published version of the manuscript.

**Funding:** This study was carried out with the support of the Cooperative Research Program for Agriculture Science & Technology Development (Project title: Technique development for manufacture and quality improvement of triticale silage; Project No. PJ01339401), Rural Development Administration, Republic of Korea. This study was supported by a 2018 Postdoctoral Fellowship Program of the National Institute of Animal Science, Rural Development Administration and Republic of Korea.

**Institutional Review Board Statement:** Not applicable.

**Data Availability Statement:** Not applicable.

**Conflicts of Interest:** The authors declare no conflict of interest.

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
