# Peer review of "Application and Future Prospective of Lactic Acid Bacteria as Natural Additives for Silage Production—A Review"

_applsci, doi:10.3390/app11178127_

Round 1

Reviewer 1 Report

The article 'Application and future prospective of lactic acid bacteria as natural additives for silage production - A review' is written in correct, clear and simple language. I believe it can be published, but the authors need to make a few small corrections: The nomenclature of acetic bacteria should be changed throughout the article - currently, for example, Lactobacillus plantarum is Lactiplantibacillus plantarum Please delete figure 4 - it adds nothing to the scientific value of the article Species name of bacteria - lowercase letter - please correct mistakes in the text

Author Response

We thank the reviewer for giving valuable comments about our research paper which is very helpful to improve the quality of the presentation of the manuscript. We have gone through the whole manuscript according to reviewer suggestions and modified the same. All changes in the manuscript have been made with red color fonts. 

Quest: The article 'Application and future prospective of lactic acid bacteria as natural additives for silage production - A review' is written in correct, clear and simple language. I believe it can be published, but the authors need to make a few small corrections: The nomenclature of acetic bacteria should be changed throughout the article - currently, for example, Lactobacillus plantarum is Lactiplantibacillus plantarum Please delete figure 4 - it adds nothing to the scientific value of the article Species name of bacteria - lowercase letter - please correct mistakes in the text.

ANS: Thank you for your positive comments on our research article. We have gone through the whole manuscript and modified the genus names according to recent guidelines (Line numbers 378-382; 399). The mistakes were also checked and corrected. Figure 4 was deleted according to the reviewer suggestion. All changes have been made with the red color font.

Reviewer 2 Report

The manuscript entitled "Application and future prospective of lactic acid bacteria as natural additives for silage production - A review" deals with an interesting topic of the use of lactic acid bacteria in silage production. 

Although the authors have done a massive effort in the literature search and review, the manuscript has some massive shortcomings.

The authors have not adequately explained the process of silage production: the initial aerobic phase, the active fermentation phase (with few stages but lasting up to 3 weeks in total, with the most intense changes in the silage material), the stable phase (minimal changes in the silage material) and feed out phase. The authors are confused about the aerobic stage - in this phase oxygen is trapped in the green mass during silage preparation, and silages are already closed. The first and most important thing in silage making is anaerobic conditions. Because of the trapped oxygen in the green mass, this part of silage making is called the first phase or aerobic phase. After the aerobic phase comes the fermentation phase (active phase) where LAB metabolize WSS (WSS generally but better is to say WSC) to lactic acid, acetic acid, propionic acid, etc., in short, lactic acid and volatile fatty acids, ethanol, and so on. And since the authors here are trying to explain the LAB use in silage production, silage production must also be explained.
Ammonia is not an acid (see line 285). Why do the authors mention ammonia with acid content? 
Why did they not explain feed out phase or aerobic stability of silages, which is a very important factor in silage use and quality? And is under the massive influence of LAB in silage production
What about the effects of LAB use on nutrient conservation; better OM preservation when inoculants are used. The manuscript does not explain changes in protein fractions or carbohydrate fractions (fiber, starch, etc.) in silages. What about WSS and buffering capacity and how they affect LAB and silage fermentation. This is very important in silage production, nutrient conservation.
Lines 105-110 New taxonomic changes to the genus Lactobacillus have been published, authors need to revise and use correct names. See http://lactobacillus.ualberta.ca/

What is the difference between homofermentative and homolactic? (See line 266) It is basically the same thing. Homofermentative LAB produces mainly lactic acid and are therefore homolactic. 

Why do you sometimes use homo-fermentative and others homofermentative?

Unfortunately, some parts of the manuscript are not adequately explained (additive usage, silage production, LAB selection for silage production, differences in homofermentative and heterofermentative LAB, etc.). 
The English language is very difficult to follow in some parts of the manuscript and there are several spelling and grammatical errors. 
Multiple-use of references. Reference 3 and 5 are the same reference, some papers are abbreviated, others are not, and so on.

Author Response

We thank the reviewer for giving valuable comments about our research paper which is very helpful to improve the quality of the presentation of the manuscript. We have gone through the whole manuscript according to reviewer suggestions and modified the same. All changes in the manuscript have been made with red color fonts. 

Quest: Although the authors have done a massive effort in the literature search and review, the manuscript has some massive shortcomings.

ANS: Thanks for your positive comments and valuable suggestion on our submitted review article.

Quest: The authors have not adequately explained the process of silage production: the initial aerobic phase, the active fermentation phase (with few stages but lasting up to 3 weeks in total, with the most intense changes in the silage material), the stable phase (minimal changes in the silage material) and feed out phase. The authors are confused about the aerobic stage - in this phase oxygen is trapped in the green mass during silage preparation, and silages are already closed. The first and most important thing in silage making is anaerobic conditions. Because of the trapped oxygen in the green mass, this part of silage making is called the first phase or aerobic phase. After the aerobic phase comes the fermentation phase (active phase) where LAB metabolize WSS (WSS generally but better is to say WSC) to lactic acid, acetic acid, propionic acid, etc., in short, lactic acid and volatile fatty acids, ethanol, and so on. And since the authors here are trying to explain the LAB use in silage production, silage production must also be explained. Ammonia is not an acid (see line 285). Why do the authors mention ammonia with acid content? Why did they not explain feed out phase or aerobic stability of silages, which is a very important factor in silage use and quality? And is under the massive influence of LAB in silage production
what about the effects of LAB use on nutrient conservation; better OM preservation when inoculants are used. The manuscript does not explain changes in protein fractions or carbohydrate fractions (fiber, starch, etc.) in silages. What about WSS and buffering capacity and how they affect LAB and silage fermentation. This is very important in silage production, nutrient conservation.
Lines 105-110. New taxonomic changes to the genus Lactobacillus have been published, authors need to revise and use correct names. See http://lactobacillus.ualberta.ca

ANS:  Thanks for your valuable comments. We have discussed phases of silage production asper reviewer suggestion (Line numbers 75-120).  

WSS is replaced by WSC in the whole manuscript

Yes we agreed with Ammonia is not an acid (see line 285) and it was corrected

We have discussed the nutrients changes in the silage treated with LAB asper  the reviewer suggestion (Line numbers 290-302). 

We have gone through the whole manuscript and modified the genus names according to recent guidelines (Line numbers 378-382; 399,)

Quest:  What is the difference between homofermentative and homolactic? (See line 266) It is basically the same thing. Homofermentative LAB produces mainly lactic acid and is therefore homolactic. 

ANS: Thanks for your valuable comments. homofermentative and homolactic both are the same and it has been modified.

Quest: Why do you sometimes use homo-fermentative and others homofermentative?

ANS: We have used homofermentative names in uniform in the whole manuscript as per suggestion.

Quest:  Unfortunately, some parts of the manuscript are not adequately explained (additive usage, silage production, LAB selection for silage production, differences in homofermentative and heterofermentative LAB, etc.). The English language is very difficult to follow in some parts of the manuscript and there are several spelling and grammatical errors. 

ANS: Thanks and sorry for the mistake and errors in the manuscript. We have gone through the whole manuscript and revised it carefully as per the reviewer suggestion. All changes were made with the red color font.

Quest:  Multiple-use of references. Reference 3 and 5 are the same reference, some papers are abbreviated, and others are not, and so on.

ANS: Thanks for your kind information. We have carefully checked deleted repeated references.

Reviewer 3 Report

In my opinion, manuscript entitled ,,Application and future prospective of lactic acid bacteria as natural additives for silage production – A review” is well written and should be considered for admission in Applied Sciences journal.

In order for the manuscript to be perfect, I suggest that the authors consider the advantage of ensiling over drying in relation to some biologically active substances, e.g. carotenoids and vitamins, which are abundant in plants. Can the authors find sources indicating what levels of these substances are lost during drying and during ensiling? Could adding lactic acid bacteria reduce the loss of these micronutrients?

Author Response

We thank the reviewer for giving valuable comments about our research paper which is very helpful to improve the quality of the presentation of the manuscript. We have gone through the whole manuscript according to reviewer suggestions and modified the same. All changes in the manuscript have been made with red color fonts. 

Quest:  In my opinion, manuscript entitled, Application and future prospective of lactic acid bacteria as natural additives for silage production – A review” is well written and should be considered for admission in Applied Sciences journal. In order for the manuscript to be perfect, I suggest that the authors consider the advantage of ensiling over drying in relation to some biologically active substances, e.g. carotenoids and vitamins, which are abundant in plants. Can the authors find sources indicating what levels of these substances are lost during drying and during ensiling? Could adding lactic acid bacteria reduce the loss of these micronutrients?

ANS: Thanks for the positive comments. We have discussed nutritional changes (water-soluble carbohydrates, organics acid profiles, plant secondary metabolites etc.) in the silages treated with LAB, Line numbers 291-306

Round 2

Reviewer 2 Report

Unfortunately, the manuscript still has major flaws and is woefully unacceptable in its current form. The ensiling process is still not properly explained and the feed-out phase is completely ignored. The aerobic phase is not properly explained (it is a first phase of ensiling when the silages are already closed, and because of the trapped oxygen we call it the aerobic phase. The aerobic phase is followed by the active fermentation phase, then the stable phase and lastly feed-out phase with the extremely important aerobic stability.

As was the case the first time, ammonia is not an acid. Homolactic and homofermentative are the same thing, references are misquoted, L. plsantarum should be L. plantarum, figures have errors (are visually inconsistent, Fig. 5 and Fig. 4), why are some mo are abbreviated, others are not in the same sentence, the manuscript lacks uniformity, no sections on the nutritional value of silage (preservation of carbohydrates, preservation of proteins in silages), what about of the parameters of good silages (LA contents, AA contents, pH, ammonia, SP, CP etc), yeasts are inhibited with AA and not fungi and etc. Unfortunately, I do not see any significant improvements in the manuscript.

Author Response

We thank the reviewer for giving valuable comments about our review paper which is very helpful to improve the quality of the presentation of the manuscript. However, the following queries have been solved in a previously submitted manuscript.  We think the reviewer used the first submitted manuscript for their review purpose.  For kind information to the valuable reviewer, we have provided the details in the comments section.

Question: Unfortunately, the manuscript still has major flaws and is woefully unacceptable in its current form. The ensiling process is still not properly explained and the feed-out phase is completely ignored. The aerobic phase is not properly explained (it is a first phase of ensiling when the silages are already closed, and because of the trapped oxygen we call it the aerobic phase. The aerobic phase is followed by the active fermentation phase, then the stable phase and lastly feed-out phase with the extremely important aerobic stability.

Answer: Thanks for your valuable comments. As per the reviewer suggestion, we have discussed the ensiling process of silage under the title of what happens to samples when they are ensiled at different stages with the necessary figure (Figure3). Please see the following details.

Line No: 75 The researchers divided the fermentation process of silages into different phases (Fig. 3) includes the initial aerobic phase, fermentation phase, the stable phase and the aerobic feed out phase [6,7]. The aerobic starts from the movement of substrates to inside the silo via the compaction process until the oxygen level are reduced [6]. This phase has the most notable feature such as the increased temperature of samples due to cellular respiration by plants and aerobic microbes. In this condition, the amount of energy is generated by respiration is much higher, this energy can be dissipated in a form of heat. With increases in acidification and concentrations of oxygen, the microorganisms that cannot survive in these conditions are inhibited, but those microbes can have survival properties that succeed. Epiphytic/ endophytic microorganisms of plants, possible contaminants or manually added inoculums are present in this stage. The most common epiphytic microbes are enterobacteria [8], yeast, moulds [9] and low numbers of LAB [10,11] with considerable diversity among plants, climates, and forage crops management.

Fermentation phase: In this stage fermentation process has accelerated by the dominant microorganism. In well-compacted silages with sufficient water-soluble carbohydrates, the LAB can dominate the fermentation process due to their adaptive and competitive growth characteristics, resulting in a well-preserved material.  The population and epiphytic diversity of LAB is highly variable and interferes with the process. In general, if LAB dominates the process, lactic acid fermentation has been started, however, may be varied the proportion of homofermentative or heterofermentative fermentation as a result of variation in lactic acid and acetic acid ratio from 1:1 to 6:1[3].  The duration of this phase or intensity of fermentation highly depends on the availability of substrates and growth conditions for dominant microorganisms.   If not LAB dominates, fail to accelerate the lactic acid fermentation and reduction in pH resulting in undesirable microbes such as clostridium, listeria and propionic acid bacteria (PAB)  started to grow and affect the silage quality[12]. Soundharrajan et al studied the impact of LAB on fermentation of Italian ryegrass at different storage periods. Reduction in pH and higher lactic acid content was noted in silage treated with LAB at 45d and 95d, further extent of storage periods affect the lactic acid fermentation due to lower LAB at 180d and 360d, resulting in lower lactic acid and higher acetic acid[13].  It may be due to nutrient availability forage samples and it favors for growth of the other microorganisms. In another study[14], rice straw treated with LAB had almost similar lactic acid and acetic acid content throughout experimental periods (45d, 90d and 365d). It confirmed fermentation process is highly based on the availability of substrates in plants. In the stable phase, the reactions are minimal, fermentation characteristics of silages were affected by storage periods. The storage time affects the pH, lactic acid, acetic acid, propionic acid and butyric acid, 1, 2-propanediol, the lactic/acetic acid ratio and NH3-N [13, 15]. All chemical and microbiological properties of the silage in the stability phase will affect the intensity of the deterioration after the silo open. When silo opens for the feed out, the fermented silages are contacted with air, which alters the conservation principles, anaerobiosis and the environmental conditions [6,16]. In this stage, the microbes that have survived in the active form or inactive spores whose growth is limited only by the oxygen availability that favor microbes grow again. Among the microbes, initially, the yeast starts the process because they are facultative anaerobes and most of the yeast has tolerance to acidity, followed by growth of mould, aerobic bacteria that result in a reduction in lactic acid, LAB population and increased pH [6,15]. These conditions can favor the growth of undesirable microorganisms that affect the silage quality.

Question: As was the case the first time, ammonia is not an acid. Homolactic and homofermentative are the same thing, references are misquoted, L. plsantarum should be L. plantarum, figures have errors (are visually inconsistent, Fig. 5 and Fig. 4), why are some mo are abbreviated, others are not in the same sentence, the manuscript lacks uniformity, no sections on the nutritional value of silage (preservation of carbohydrates, preservation of proteins in silages), what about of the parameters of good silages (LA contents, AA contents, pH, ammonia, SP, CP etc), yeasts are inhibited with AA and not fungi and etc. Unfortunately, I do not see any significant improvements in the manuscript.

Answer: Thanks, and sorry for the intervention, we think the reviewer used the initially submitted manuscript for their review purpose. When first revision, we deleted issued on ammonia is not an acid, some figure and upgraded this review paper via including changes of plant metabolites, water-soluble carbohydrates and other nutrient profiles changes after the ensiling process. For your kind information: Line No: 289 The addition of LAB during ensiling process improved fermentation quality (higher lactic acid), and maintained the crude protein (CP), acid detergent fiber (ADF), and neutral detergent fiber contents (NDF)[13,14]. Another report claimed that the addition of LAB significantly increased nutritive profiles of silage at different storage periods (Wang et al., 2019). Increased acid detergent lignin[73] and dry matter content (DM)[74] were noted in silages treated with LAB, but the DM level was varied at different storage periods [61]. The organic matter (OM) and DM content was increased in the fresh and rain treated ryegrass silages treated with LAB [75]). Water-soluble carbohydrates (WSC) level was reduced in the silage treated with LAB [73,74]; it may be due to LAB being able to utilize WSC and converts into organic acids. The plant metabolites such 3-hydroxydecanoic acid, 2-hydroxy-4-methylpentanoic acid, benzoic acid, catechol, hydrocinnamic acid, salicylic acid, 3-phenyllactic acid, 4-hydroxybenzoic acid, (trans, trans)-3,4-dihydroxycyclohexane- 1-carboxylic acid, p-hydrocoumaric acid, vanillic acid, azelaic acid, hydroferulic acid, p-coumaric  acid, hydrocaffeic acid, ferulic acid, and caffeic acid were increased in grass silage inoculated with LAB[76]. LAB increased α-tocopheraol level in silage prepared from the mixture of birds foot trefoil and timothy, red clove and meadow fescue, red clover and timothy whereas β-carotene was level either same or slightly reduced in same silage mixture [77], whereas, the silages from rye had higher β-carotene  [78]